# Role of Marine Bacterial Contaminants in Histamine Formation in Seafood Products: A Review

**DOI:** 10.3390/microorganisms10061197

**Published:** 2022-06-11

**Authors:** Adnorita Fandah Oktariani, Yan Ramona, Putu Eka Sudaryatma, Ida Ayu Mirah Meliana Dewi, Kalidas Shetty

**Affiliations:** 1Doctoral Study Program of Biology, Faculty of Mathematics and Natural Sciences, Udayana University, Denpasar 80361, Bali, Indonesia; adnoritafandah@student.unud.ac.id; 2PT. Intimas Surya, Denpasar 80222, Bali, Indonesia; 3Integrated Laboratory for Biosciences and Biotechnology, Udayana University, Denpasar 80361, Bali, Indonesia; 4Fish Quarantine Inspection Agency, Denpasar 80361, Bali, Indonesia; putueka.sudaryatma@gmail.com; 5School of Biology, Faculty of Mathematics and Natural Sciences, Udayana University, Denpasar 80361, Bali, Indonesia; mirahmeliana@gmail.com; 6Department of Plant Sciences, North Dakota State University, Fargo, ND 58102, USA

**Keywords:** biogenic amine, histamine producing bacteria (HPB), marine bacteria

## Abstract

Histamine is a toxic biogenic amine commonly found in seafood products or their derivatives. This metabolite is produced by histamine-producing bacteria (HPB) such as *Proteus vulgaris*, *P. mirabilis*, *Enterobacter aerogenes*, *E. cloacae*, *Serratia fonticola*, *S. liquefaciens*, *Citrobacter freundii*, *C. braakii*, *Clostridium* spp., *Raoultella planticola*, *R. ornithinolytica*, *Vibrio alginolyticus*, *V. parahaemolyticus*, *V. olivaceus*, *Acinetobacter lowffi*, *Plesiomonas shigelloides*, *Pseudomonas putida*, *P. fluorescens*, *Aeromonas* spp., *Photobacterium damselae*, *P. phosphoreum*, *P. leiognathi*, *P. iliopiscarium*, *P. kishitanii*, and *P. aquimaris.* In this review, the role of these bacteria in histamine production in fish and seafood products with consequences for human food poisoning following consumption are discussed. In addition, methods to control their activity in countering histamine production are proposed.

## 1. Introduction

Biogenic amines (BA) are organic compounds with low molecular weight [1]. Their formation in seafood products is mainly catalyzed by bacterial histidine decarboxylase (HDC), which is an enzyme that is associated with seafood spoilage using free histidine as its main substrate [2,3,4]. The formation of BAs in seafood products has been a significant public health concern due to their physiological and toxicological effects [5]. Several BAs found in foods include histamine, tyramine, phenylethylamine, putrescine, agmatine, cadaverine, spermine, spermidine, and tryptamine. Among these allergenic compounds, histamine has been reported to be the most frequently found BA in foods such as fish, fish products, fermented meats, vegetables, and dairy products as well as alcoholic beverages [4,5,6].

Histamine with the molecular formula and molecular weight of C_5_H_9_N_3_ and 111.14 g/mol [3,7], respectively, has caused many a scombroid poisoning and has been considered as a foodborne chemical hazard. Exposure of seafood products to histamine-producing bacteria at high ambient temperatures leads to the accumulation of histamine in such products [8]. The histidine decarboxylase enzyme released by these bacterial contaminants leads to decarboxylation of free histidine to form histamine in seafoods [9]. Contamination of seafood products by histamine-producing bacteria commonly occurs following their harvesting or during post-harvest handling of the products. A high concentration of histamine is frequently detected in fish species belonging to the family of Scombroidae [10,11,12,13]. Due to this, the food poisoning phenomenon caused by fish species within this family is referred to as scombroid poisoning [14]. Several species of non scombroid fish such as mahi-mahi, anchovy, amberjack, marlin, bluefish, herring, and sardine may also be implicated in scombroid poisoning. Furthermore, fish species with dark muscles such as tuna, blue scad, chub mackerel, bonito, and saury may also contain high levels of histamine due to the high free histidine level in their muscle tissues [3,4,15,16,17,18,19,20,21].

The allergenic effects of scombroid toxin (histamine) from fish poisoning usually appear a few minutes to a few hours after a person has ingested histamine-containing fish and affects various organ systems, which lead to the occurrence of various intestinal and extra-intestinal symptoms [21,22,23]. These symptoms include mild illness such as rash, urticaria, nausea, vomiting, diarrhea, flushing, and headache with common symptoms of tingling and itching of the skin. However, if foods containing high levels of histamine are ingested, enzymes produced by the epithelial cells of our intestine may fail to function to cope with this poisoning, and some of this histamine may be absorbed into our circulatory system and stimulate several symptoms such as neurological, gastrointestinal as well as respiratory symptoms to occur [3,4,6,7,21,22]. The period of histamine poisoning symptoms may vary among individuals, and commonly last for one or several days [24]. This depends on the level of histamine intake and the degree of the patient’s sensitivity [4,7,15].

In the case of skin diseases such as psoriasis due to incremental increase in the histamine level, this BA may function as a pleiotropic mediator [25]. In our body, mast cells and basophils are cells involved in our immune system that produce the majority of histamine [26]. High levels of histamine are stored in these cells and are released quickly in response to immune system stimulation [27]. Basophile-released histamine found in histamine allergenic associated patients is often used for diagnostic purposes (histamine release test) [28]. Secretion of this allergenic compound by many innate and adaptive immune cells occurs after decarboxylation of the free histidine by HDC [29].

The purpose of this review is to provide integrated information and elaborate the role of marine bacteria in the formation of histamine (a biogenic agent causing food poisoning cases) in seafood products. The article also highlights some methods to reduce or eliminate histamine production in seafood products, so that poisoning following seafood product consumption can be avoided.

## 2. Histamine Responses in the Human Body

Many acute symptoms resulting from allergenic responses may involve the role of histamine as a mediator [30]. Its long-term immune response regulation has also been well documented. There are four types of histamine receptors (H1R, H2R, H3R, and H4R) through which this histamine signaling determines the triggered responses [31]. In mediating hypersensitive and allergenic responses, H1R and HR2 work antagonistically. Being a neurotransmitter, the role of H3R is to release the control of the presynaptic receptor. This works at the presynaptic membrane of neurons in the central nervous system and prevents the histamine from reaching neural and other neurotransmitters. Receptor H4R is important in the production of cytokines by innate immunity associated cells (eosinophils, mast cells, basophils, and dendritic cells (DCs) as well as T cells [29,32]). Smooth muscle contraction, permeability of the vascular system, and the ability of our body to respond to allergans involve histamine actions through processes predominantly facilitated by H1R. Many of these processes require a high-affinity binding of Immunoglobulin E (IgE) to its specific receptor. Once activated, the mast cells and basophils will degranulate and mediate the release of histamine. In the gastric mucosa, histamine is released by enterochromaffin-like cells in the form of paracrine functions to control the secretion of gastric acid and respond to hormonal and neural stimuli. This process is basically via the H2R. Histamine can have mutual collaboration with other transmitters of the central nervous system (CNS), and in such cases, the histamine effects are mediated by H1R, H2R, and H3R signaling [33]. The H3Rs in the form of presynaptic autoreceptors prevent histamine synthesis and its release in the histaminergic neurons [34]. They also function as postsynaptic heteroreceptors, which are distributed in the central nervous system. Both presynaptic and postsynaptic heteroreceptors have implications in the basic homeostatis and in the improvement in brain functions. Finally, eosinophil chemotaxis and selective mast cell recruitment are recently known to be controlled by H4R. These mechanisms result in the amplification of histamine-mediated immune responses followed by chronic inflammation [35]. In addition, it modulates dendritic cell activation and helper T cell (TH) differentiation, and is thus characterized as part of the overall immune system. This indicates that the receptors of histamine play a significant role as immunomodulators in atopic and inflammatory pathologies [36].

The effects of histamine in the respiratory, cardiovascular, gastrointestinal, hematological, or immunological systems, and the skin will be initiated following its binding to its receptors located on the cellular membrane [27,37,38,39,40]. Under normal conditions in our body, the rapid detoxification of exogenous amines absorbed from food is catalyzed by amine oxidase type enzymes or by conjugation [41]. However, when a patient consumes a high level of histamine, the histamine detoxification process may be slowed, resulting in histamine accumulation in the patient’s body [42]. Enzymes, in addition to monoamine oxidase (which includes diamine oxidase and histamine N-methyltransferase), have been extensively studied for their ability to modulate histamine detoxification [43].

The role of diamine oxidase enzyme in the detoxification of histamine has been extensively reviewed by Devivilla et al. [44]. In this review, it was mentioned that an increase in histamine level in the blood plasma because of decreased activity of the diamine oxidase enzyme may lead to many symptoms that affect various organ systems. Interestingly, some lactic acid bacteria such as *Lactobacillus reuteri*, *L. casei*, and *L. delbrueckii* subsp. *bulgaricus*, which have been recently considered as potential inhabitants of the human intestinal tract or those developed as probiotic candidates in recent years, are reported to have the capability to produce the HDC enzyme, and therefore are able to accumulate histamine in the intestine. This biogenic agent has the potential to enter the blood streamand subsequently result in increased blood histamine concentration, which subsequently promotes histamine sensitivity in some individuals [22].

Reactions to histamine among people may vary, and this has been revealed in many studies that correlated with the microbial composition in their intestinal tract [29,45]. Some studies reported that the intestinal microbial composition of people with histamine intolerance was found to be significantly different when compared to those healthy and food-allergenic individuals. The differences were found to be insignificant in many cases [46]. Bacteria that belong to the genus *Butyricimonas*, however, were detected to have lower cell density in people with histamine intolerance when compared to food hypersensitive and food allergenic individuals [22].

An elevated proportion of *Proteobacteria* that leads to low-grade intestinal inflammation is hypothesized to be due to dysregulation of the innate immune response [47]. This inflammation may lead to dysfunction of the intestinal epithelial tissue and elevation of oxygen levels in the colon [48], which results in the growth promotion of facultative anaerobic bacteria (e.g., various species of *Proteobacteria*). These facultative anaerobic bacteria will compete with obligate anaerobic bacteria including those beneficial for human health (those within the group of *Bifidobacteria*) [49]. *Proteobacteria* abundance has been characterized in healthy persons as being between 2.5 and 4.6%, but it increased by 1.3% to 34.6% in patients with a histamine intolerance. This implies dysbiosis and/or impaired epithelial function in this patient population. Interestingly, some studies have reported that a rise in the number of Gram-negative *Morganella morganii* is an indication of dysbiosis. This species belongs to the family of *Enterobacteriaceae* and is part of the phylum *Proteobacteria*. Furthermore, metabolic dysfunction mediated by commensal microbes in the normal human intestinal tract have also been reported to play a significant role in the development of dysbiosis [22,36].

## 3. Marine Bacteria with the Capability to Produce Histamine

Marine bacteria with the capability to produce HDC are a serious concern in seafood industries as they can accumulate histamine in seafood products through the conversion of free amino acid histidine into histamine [50,51]. In the family of Crustaceae, this biogenic amine has been reported to be produced naturally along with the ecology of the associated bacteria, and it acts as a neurotransmitter [52]. HDC producing marine bacteria commonly found in seafood products include both Gram-positive and Gram-negative bacteria. Fish and other seafood products are commonly contaminated by Gram-negative bacteria of enteric and marine origin with the capability to produce histamine. Gram-positive histamine producing bacteria, on the other hand, are commonly detected in fermented foods such as cheese, wine, and beer [15,53]. Decarboxylase enzymes capable of utilizing histidine, lysine, and ornithine produced by bacteria belonging to the *Enterobacteriaceae* generally have high activity in converting these free amino acids into various biogenic amines [54]. Genera of *Morganella*, *Enterobacter*, *Hafnia*, *Proteus*, and *Photobacterium* as well as different pseudomonads and lactic acid bacteria within the genera of *Lactobacillus* and *Enterococcus* have been widely reported to have the capability to produce the HDC enzyme [3,16,23,53].

In fish or fish products, bacterial species including *Proteus vulgaris*, *Proteus mirabilis*, *Enterobacter aerogenes*, *Enterobacter cloacae*, *Serratia fonticola*, *Serratia liquefaciens*, *Citrobacter freundii*, *Citrobacter braakii*, *Clostridium* spp., *Raoultella planticola*, *Raoultella ornithinolytica*, *Vibrio alginolyticus*, *Vibrio parahaemolyticus*, *Vibrio olivaceus*, *Acinetobacter lowffi*, *Plesiomonas shigelloides*, *Pseudomonas putida*, *Pseudomonas fluorescens*, *Aeromonas* spp., *Photobacteriumdamselae*, *Photobacterium phosphoreum*, *Photobacterium leiognathi*, *Photobacterium iliopiscarium*, *Photobacterium kishitanii* and *Photobacterium aquimaris* have been successfully isolated and confirmed to have the capability to produce histamine [3,7,17,53,55,56]. In fermented foods such as cheese, *Lactobacillus parabuchneri*, *Lactobacillus paracasei* and *Pediococcus pentosaceus* (LAB) have been reported as histamine producers [5]. In Indian anchovy, mackerel, and filleted tuna [7,16,21], *Morganella morganii* has been reported as an important histamine-producing bacterium, and its wide prevalence makes it a strong candidate in cases of histamine poisoning. Based on the histamine level produced under ideal conditions (at least at 15 °C and 48 h incubation) either in tuna fish infusion broth or in trypticase soy broth (TSB) medium with 1 to 2% histidine in it, histamine producing bacteria (HPB) were divided into two categories. HPBs with the capability to produce histamine at a higher level than 1000 mg/L were categorized as high-HPBs, while those with the capability to produce such compounds at a lower level than 500 mg/L were categorized as low-HPBs. *M. morganii*, *M. psychrotolerans*, *Klebsiella pneumoniae*, *K. oxytoca*, *E. aerogenes*, *E. cloacae*, *R. planticola*, *R. ornithinolytica*, *Clostridium perfringens*, *P. damselae*, *P. kishitanii*, and *P. angustum* are examples of high-HPB and they are all identified as prolific histamine forming bacteria capable of producing more than 1000 mg/L histamine in broth culture. On the other hand, *Hafnia alvei*, *Vibrio alginolyticus*, *Escherichia coli* and *Citrobacter freudii* are examples of low-HPB, with ability to produce histamine less than 500 mg/L [4,17,20,21,57,58]. Among the HPB, *M. morganii* was reported by Roy and Nayak [16] to have the capability to produce the highest level of histamine (2126 mg/L). Other species such as *K. variicola*, *Proteus vulgaris*, *P. penneri*, *P. mirabilis*, *Providencia rustigianii*, *A. faecalis*, and *Psychrobacter pulmonis* have the capability to produce histamine at levels of 1822 mg/L, 1738 mg/L, 1440 mg/L, 1253 mg/L, 1011 mg/L, 508 mg/L, and 471 mg/L, respectively.

HPB can be further classified into terrestrial and marine bacteria based on their habitat. Terrestrial histamine-producing bacteria include *Morganella* and *Enterobacteriaceae*. These both originate from the intestinal tract of hot-blooded animals [20,59]. These bacteria primarily proliferate at moderate temperatures. Meanwhile, marine HPB include those belonging to the genera of *Vibrio* and *Photobacterium* and are naturally seawater inhabitants. Those that proliferate at moderate to low temperatures are also included in this group [7]. *Photobacterium* sp. has been suggested to be responsible for producing histamine in psychrophilic conditions. Several bacteria with the ability to survive at low temperature and salty water and commonly found as normal microbiota of marine fish species have been reported to produce a high level of histamine at the incubation temperature of 2.5 °C [19]. Such types of bacterial species are often associated with the gills, intestinal tract, or skin of marine fish species. Therefore, if the harvested fish are stored at temperatures higher than 15°C for several hours, then a high concentration of histamine will be detected due to the activity of HDC enzymes produced by the HPB contaminants. Such conditions allow spoilage microorganisms including species of *Morganella*, *Enterobacter*, *Hafnia*, *Raoultella*, and *Photobacterium* to rapidly proliferate and produce histamine in the fish [4].

*Raoultella**ornithinolytica*, a prolific histamine producer isolated from mahi-mahi fish (*Coryphaena**hippurus*) products, was found to produce histamine at a rate of >500 mg/L in a broth culture incubated at 35 °C for 24 h. The rate of histamine formation by this bacterium was found to significantly increase when incubation of the broth culture was prolonged to 48 h at temperatures of 25 °C and 37 °C. When the incubation temperature was lowered to 15 °C, the rate of histamine production was found to decrease to 38 mg/100 g following 24 h of incubation. Long-term incubation of this culture for up to 96 h resulted in a 400% increase in the histamine level (165 mg/100 g). This indicated that the temperature and incubation time determined the histamine level produced by *R. ornithinolytica* in the fish sample, where 37°C was reported to be the optimum temperature for this species to produce histamine [3]. 

According to [60] two strains of *R. ornithinolytica* has frequently been found to contaminate tuna, albacore, and sailfish, and are able to produce histamine in TSB with 1.0% L-histidine in it at the rates in the range of 276.6 ppm to 561.8 ppm. Another prolific histamine producer (*E. aerogenes*) was also found as a contaminant of these seafood products. In recent years, these two prolific histamine producers have been isolated from milkfish stick and dumpling, dried milkfish, and tuna dumplings. They are reported to be potential species with the ability to produce >400 ppm of histamine in TSBH. As fish and other seafood products can easily be contaminated by HPB, these products can then be considered as vehicles for histamine to cause histamine poisoning related cases [60].

As histamine production and accumulation normally rapidly begins following 12 h of storage at 25 °C, temperature control should be applied to reduce the rate of histamine production in seafood products. The freezing process was found to be effective to control *R. ornithinolytica* from producing histamine in the products. In an experiment conducted by [3], *R. ornithinolytica* was reported to show rapid growth and produced histamine in the fish muscles when the temperature was increased to 25 °C from the frozen condition. This indicates that this bacterial species becomes dormant under the frozen condition and starts to recover when the temperature is increased to its optimal growth temperature. During its recovery, this species will start to produce histamine in the products. Similar temperature effects on histamine formation were also reported by [60], who found that the formation of histamine in the milkfish dumpling samples was significantly faster at 25 °C and 37 °C than at 15 °C and 4 °C. Histamine accumulation often occurs when frozen products are thawed and left at room temperature for a certain period of time. Histamine is a heat resistant toxin, and therefore its toxicity remains intact in canned or cooked fish products [60].

Crustaceans are also susceptible products for contamination by HPB as they contain a high level of free amino acids [8,61]. Due to its relatively high moisture content, this seafood becomes perishable due to contamination by microbes including those with the capability to produce histamine. High levels of bacteria are commonly found in the intestinal tract of crayfish, running along the dorsal side of this species. These bacteria play a significant role in the deterioration of this seafood product and lead to the formation of biogenic amines during storage. The accumulation of histamine in this product to levels exceeding the safety limit [52] will be dependent on the bacterial community composition, storage time, and incubation temperature.

The histamine levels in crayfish tissue were shown to rise over time in refrigerated storage, where they were exposed for 14 days to copper or copper-free water and the histamine levels were greater in the copper-free water. Copper was discovered to diminish the microbial diversity in water and to alter the bacterial community structure in crayfish guts. Many bacteria, including *Pseudomonas syringae*, *Xanthomonas campestris* pv. Juglandis, and *E. coli* strains, are resistant to copper exposure. However, *Pseudomonas* species (*P. fluorescens*, *P. putida*, and non-fluorescent *Pseudomonas* spp.) contribute to the reduced histamine levels [52].

HDCs are classified into two types: those whose activity is dependent on pyridoxal-5′-phosphate as a cofactor and those whose activity is dependent on pyruvoyl (covalently bonded pyruvoyl moiety) [62]. Free histidine is required for histamine generation by HPB and is carried into the cytoplasm of the bacterial cells, where histamine production through decarboxylation occurs. Histamine is then transferred out of the cell after it has been produced [63]. Therefore an antiporter is needed for this process to occur [64]. Various Gram-negative bacteria create histamine in the presence of pyridoxal phosphate, while Gram-positive bacteria, notably lactic acid bacteria engaged in food fermentations or as food contamination, usually make histamine in the presence of pyruvoyl moiety. Amino acid decarboxylase enzymes are required to produce histamine by the decarboxylation of free amino acids. The majority of these enzymes need pyridoxal 5′-phosphate as an important cofactor [19].

Mou et al. [65] found six phyla including *Proteobacteria*, *Actinobacteriota*, *Bacteroidota*, *Firmicutes*, *Fusobacteriota*, and *Verrucomicrobiota* as bacteria to be putatively secreting histamine. Among these phyla, only *Proteobacteria* species were found to be capable of synthesizing pyridoxal 5′-phosphate or (PLP) dependent HDC [66]. Pyridoxal-P-dependent HDC and pyruvoyl-dependent HDC enzymes produced by groups of *Morganella* and *Lactobacillus*, respectively, have been reported to have similar efficiencies in the decarboxylation of histidine, although their molecular organization and substrate specificity are different [19]. The pyridoxal-P enzyme and the formation of biofilm in the fish processing industries by some bacterial communities have become big concerns in recent years [67]. This biofilm consists of extracellular polymers known as polysaccharide intracellular adhesive substances. The existence of these substances make these bacterial communities become more resistant than their planktonic form. Within this biofilm structure, bacteria may thrive on contact surfaces for long periods and pose a risk of the persistent contamination of fish at various processing stages. *Morganella* spp. with flagella and urease activity is an important histamine producer and has become the main determinant of the colonization and formation of biofilms [16].

Roy and Nayak [16] noted that the exposure of sodium hypochlorite at the rate of 1 and 3 ppm to the HPB accelerated the biofilm formation by these bacteria. *K. variicola* was an exception in this study where its ability to form a biofilm was found to be significantly decreased following the exposure to 5 ppm sodium hypochlorite. Biofilm formation can be inhibited by exposure to high chlorine concentrations. Chorine exposure will lead to the disintegration of the biofilm architecture followed by the shedding of bacterial cells. Some species of HPB show resistance to low levels of chlorine. This suggests that the accumulation of histamine in the fish or processed fish will occur as the HDC producers within the biofilm on the fish or processed fish surface will continuously secrete histamine in these products [56,68].

## 4. Detection of Histidine Decarboxylase Cluster Gene

HPBs in foods are commonly detected by applying the culture method in Niven’s medium supplemented with pH indicators to indicate a rise in pH due to histamine formation from free histidine in the medium. This medium is good for presumptive tests and is widely used to screen suspected HPB, although in many cases, this medium has been reported to produce false positive results, and therefore further tests are needed for confirmation. Additionally, growth inhibition by some HPB often occurs due to the low pH of the medium (pH 5.3). This leads to a false negative [21].

Sequencing of 16S rDNA with *gyrB* and *hdcA* as the target genes has been suggested to be one of the most accurate methods to identify HPB within the group of Gram-negative bacteria. Other available methods used to detect such bacterial groups include single-strand conformation polymorphism (SSCP) and high-resolution melting analysis (HRMA). SSCP is a rapid screening method that requires high-resolution electrophoresis, while HRMA is a method that detects differences in the base sequences of PCR products based on the variation of the melting temperature of double-stranded DNA related to differences in the base sequences. In recent years, PCR and HRMA, in conjunction with the designed primers, have been applied to categorize 20 strains of HPB into three groups based on differences in their melting temperature (Tm) values [7].

The identification of isolates with the capability to produce histamine was achieved by amplification followed by the sequencing of approximately 1400 bp of their 16S ribosomal DNA (rDNA) or 16sRNA [17,52,60]. In this amplification, the *hdc* gene of the Gram-negative bacteria is the target for amplification using primers of hdc-f and hdc-r with the capability to amplify a DNA fragment of 709 base pairs. Application of such primers (hdc- f/hdc-r) in the polymerase chain reaction (PCR) to all but one (*Citrobacter brakki*) Gram-negative histamine-producer bacteria could be rapidly detected [19]. Meanwhile, the *hdc* gene of Gram-positive bacteria is usually amplified using HDC-3 and HDC-4 primers. These primers could amplify the DNA fragment of 435 bp [52]. The gene of *hdcA* (the internal parts of *hdc* gene cluster) is commonly detected using two sets of different primers (Hdc1/Hdc2 and JV16HC/JV17HC) in the amplification of such genes [5]. In addition, some multiplex PCR reactions have been developed to achieve the simultaneous amplification of the gene encoding production of various amino acids within the enzymes of the group of decarboxylases including those encoding for HDCs [19].

The contamination of seafood products by HPB results in an increase in the histamine value in these seafood products. The level of this allergenic compound needs to be regularly assayed to avoid histamine poisoning. Currently, there are two commonly used assays in the quantification of histamine levels in biological samples. These include enzyme-linked immunosorbent assay (ELISA) and the colorimetric enzymatic assay using histamine dehydrogenase with the approximate detection limits of 0.1 µM and 0.5 µM, respectively. These methods have been found to be very effective, but they tend to be costly. Sometimes, they require careful storage and handling mechanisms. To avoid such disadvantages, the use of methods such as high-performance liquid chromatography (HPLC), mass spectrometry, and surface-enhanced Raman spectroscopy (SERS) have recently been targeted. These instrumentation analyses offer advantages such as high sensitivity, although such assays are time-consuming and require sophisticated instruments and skilled operators [69]. Small molecular probes for histamine analysis have also been reported, but such methods in the field still have limited applications [28].

Molecular-based probes were used in combination with colony lift hybridization methods to quantify Gram-negative HPB in scombroid fish by virtue of the specificity of a probe to target the *hdc* gene of these organisms. When applied to screen individually by dot blot hybridization against each of the 152 isolates in the strain bank, all probes failed to detect these 152 histamine-producing isolates [70]. However, when the 709 bp probes were mixed in equal proportion (1:1:1:1), this probe cocktail was able to detect all 73 of the high histamine producing strains (1000 ppm), but failed to detect strains with a low level of histamine production (126–500 ppm). Overall, the *hdc*-probe mix used in the study was able to discriminate between high producers and non-producers, but all low histamine producing strains are designated as non-producers using this probe mix [21].

Much attention has been given to the characterization of bacterial genes involved in the regulation of histamine production [71,72]. The capacity to sequence such genes would enable the ability to create and improve novel and efficient approaches to detect HPB in the samples. The genes involved for histamine production in Gram-positive bacteria are clustered into a cluster known as the HDC cluster. Three genes that are normally oriented in the same direction (*hdcP*, *hdcA*, and *hdcB* genes) have been identified as being responsible for histamine production in HDC cluster-based histamine-producing bacteria. The secretion of histamine from the cytosol in exchange for histidine is undertaken by the antiporter *hdcP*. These proteins are encoded by the genes *hdcA* and *hdcP* [73]. The HDC cluster has been described in several lactic acid bacterial species such as *Lactobacillus saerimneri*, *L. parabuchneri*, *L. hilgardii*, *Tetragenococcus halophilus*, *Streptococcus thermophiles*, *L. reuteri*, *Tetragenococcus muriaticus*, and *Lactobacillus vaginalis* [6].

Although the *hdc* genes in Gram-negative bacteria have been reported to evolve from a common ancestral gene and share extensive similarities among them, it is always possible to sequence the variations among the *hdc* genes of the Gram-negative bacteria with the capability of producing either a high or low level of histamine. The results, however, may sometimes result in false negatives [16].

## 5. Factors Affecting Production of Histamine by Bacteria

Histamine production by HPB is affected by many factors such as pH, temperature, substrate concentration, and many other environmental factors [74]. A study on *L. vaginalis* grown at 37 °C for 14.5 h and in the absence or in the presence of histidine showed that no histamine was detected in the absence of histidine, but histamine started to appear following the addition of histidine at a concentration of 0.1 mM. The histamine level was found to increase in line with the increments of histidine and reached its peak at 5 mM histidine [75,76]. This indicates that the activity of HDC produced by the relevant bacterial species is significantly affected by the substrate concentration. It was also extensively reviewed by Rossi et al. [77] that free histidine affects the expression of *hdcA*, and its expression is positively correlated with the histidine concentration. In such cases, histidine can be considered as an inducer for this gene to be transcribed, enabling the HPB to utilize and convert free histidine into histamine [78]. Furthermore, it was also reported that the induction of the *hdcA* gene was enhanced up to a histidine concentration of 0.5 mM, after which no significant increase was observed [79]. The same phenomenon was also observed in the quantification of *hdcP* expression using the same cDNAs where the histidine concentration at 0.5 mM was observed to be the optimum concentration to induce these genes in relation to histamine production [80].

Interactions between HPB and their hosts may be mediated by histamine [81,82]. In other words, histamine can play an important role as a central signal molecule in this interaction. In many types of Gram-negative bacteria, histamine production is encoded by the *HDC* gene [63]. A cluster of *HDC* consists of two super families, namely Gram-negative bacteria, which need a coenzyme in the form of pyridoxal phosphate, and Gram-positive bacterial species, which require a covalently bound pyruvate moiety for catalysis. When inoculated in a medium containing histamine as the sole carbon and nitrogen source, a strain of *P. aeruginosa* (PAO1) showed excellent growth response [83]. This indicates that such strains harbor pathways functioning in the degradation of histamine. This histamine catabolism involves the regulation of the *hinK* gene, which is responsible in such strains for histamine degradation. Regulation of the *hinK* gene in histamine catabolism is commonly coupled with transcriptional regulators of *HinB* and *HinJ*. To transport the histamine into the cell cytoplasm, an APC (amino-acid-polyamine-organo cation)-type transporter is required, which involves the expression of the gene *HinA* [81]. 

Histamine levels in fish muscles have been shown to be affected by different stages of microbial development. *E. aerogenes* and *M. morganii* are two bacterial species that exhibit varying rates of histamine synthesis at various stages of development. Kim et al. [84] discovered that *M. morganii* created the most histamine in milkfish muscle during the late exponential phase of its development, while the former produced the most histamine during the stationary period of its growth. Furthermore, the rate of histamine generation in the fish samples was determined by the species of HPB and fish species [58,60].

## 6. Anti-Histamine Compounds and Solutions

Histamine when produced at certain levels in seafood products by *Staphylococcus aureus*, *Bacillus subtilis*, *Salmonella typhi* and *E. coli* [85,86] may have a poisoning effect in humans. Many efforts have been pursued to inhibit the histamine production in seafood products. The *Polyscias guilfoylei* leaf methanol extract (PGE), for example, was found to inhibit the activity of *E. coli* to release histamine when applied at its minimum inhibitory concentration (9.76 μg/mL) [85]. Compounds that block the allergenic activity of histamine are called anti-histamine compounds [87]. The existence of such compounds in the presence of histamine producing bacteria will improve the effectiveness of antibiotics to control histamine producers [88]. Mepyramine, for example, was found to improve the efficacy of several antibiotics such as amoxicillin and sulfadiazine when combined to control the growth of *E. coli* ATCC^®^ 25922™ and *E. coli* PIG 01, although the mechanisms by which such anti-histamines improve the efficacy of such antibiotics are still inconclusive [86]. According to El-Banna et al. [89], the improved effectiveness of such antibiotics to control histamine producing bacteria in the presence of histamine could be due to anti-histamines being inhibitory to the bacterial efflux pumps. Another possibility of this phenomenon may be due to the main structural feature of anti-histamines, which is likely to be similar to surfactants [90]. With this structure, it might cause alterations in the biological membrane permeability of the histamine producers [86].

Clemastine, ranitidine, and FR145715 are other examples of compounds considered as anti-histamines [91]. Clemastine was found to inhibit the blood and liver stages of the parasites *Plasmodium falciparum* and *P. berghei*, respectively [92]. Clemastineat 10 μM was found to totally inhibit parasite loads in both the blood and liver cells [92]. Meanwhile, FR145715 (which is considered as a novel anti-histamine) and ranitidine are two anti-histamines with potent H2 receptor antagonist activity. In recent years, the pharmacological profile and efficacy of FR145715 and ranitidine as anti-ulcer agents have been compared in many studies [93,94]. Their effectiveness to antagonize histamine-induced positive chronotropic response in isolated guinea-pig atrium has also been compared [95]. It was reported in these studies that FR145715 was found to be approximately 300% more effective than ranitidine as a potent histamine H2 receptor antagonist. However, FR145715 was found to be approximately 6 and 16 times less effective in histamine-stimulated acid secretion when applied intravenously and intraduodenally, respectively, in Schild’s rats. The effect of FR145715 to inhibit acid secretion was not in line with this result when intraduodenal administration was applied in a study of Shay’s rats. In this study, FR145715 was found to spontaneously inhibit acid secretion with slightly higher potency when compared to ranitidine [96,97]. The differences in the effects shown by these two anti-histamines might be due to the differences in the pharmacokinetics of these two drugs when applied under different rat conditions (conscious Shay’s rats and anaesthetized Schild’s rats) [96]. Anti-histamine FR145715 was also found to be more effective against *H. pylori* when compared to conventional histamine H_2_ receptor antagonists (ranitidine, cimetidine, famotidine, and roxatidine) [97].

## 7. Histamine Control in Seafood Products

The control of histamine in seafood products has relied mainly on the time and temperature controls during harvesting, handling, processing, and storage [98]. Histamine has been reported as a thermostable or heat resistant toxic compound, and therefore once it occurs in the seafood product, it is impossible to destroy it in the product [4]. Its level may even increase during cooking [4,15,28,50]. Among the HPB, *M. morganii* was found to be the most heat-resistant bacteria in contaminated tuna loins when compared to other species (*E. aerogenes*, *H. alvei*, and *R. planticola)* [4,15]. As soon as fish mortality occurs, histamine formation will start via decarboxylation catalyzed by HDC produced by the relevant bacterial species, although the products are stored at a temperature above 4°C for an extended period. Once excreted into the seafood products, HDC will continuously convert free histidine into histamine, eventhough the relevant bacteria are potentially inactivated [20]. The destruction of HDC in tuna depends on the time of heat exposure and the degree of heating. Heating the seafood products during cooking to 60°C may not be fully effective to inactivate the HDC enzyme of *E. aerogenes* if the exposure time is too short. Therefore, an appropriate temperature treatment and the duration of heat exposure during cooking or storage need to be determined and applied to prevent histamine formation as well as destroy the already existing histamine in the seafood products [15].

The processing of previously frozen fish that include thawing, precooking, cleaning, and the packaging of such seafood products are generally undertaken at temperatures of higher than 21 °C for up to 12 h, according to manufacturer’s guidelines [99]. However, the completion of some steps of this process may need more time. To cope with this problem, the precooking method of the seafood products has been applied by tuna producers to reduce the processing times [15]. Histamine accumulation commonly occurs during the prolonged interval of time between refrigeration and the precooking steps [4].

In traditional methods, chilling and freezing [17] have generally been applied to prevent histamine accumulation in seafood products. Rapid chilling of seafood products following harvesting to prevent histamine accumulation in such products has also been recommended by the U.S. Food and Drug Administration (FDA), and this method is extensively explained in the guidance issued by the FDA. This will limit the scombrotoxin-forming fish from exposure to temperatures higher than 4.4 °C throughout the cold chain, although such methods cannot avoid the formation of histamine and other biogenic amines as many HPB are still active under psychrotrophic conditions [99]. This indicates that histamine formation in seafood products can only be controlled through the inhibition of HPB growth and inhibiting the activity of HDC enzymes and/or degradation of histamine [100,101]. Borriello et al. [102] suggested a promising method to control histamine accumulation by adjusting the pH of the products.

In addition to the above proposed methods, the application of trisodium phosphate (TSP) has been practiced by the food industries. In this application, the seafood products are dipped or sprayed with TSP to increase the surface pH of the products up to approximately pH 12. The effectiveness of TSP application to prevent histamine formation was found to increase in mahi-mahi and tuna samples when coupled with treatment in vacuumed packaging. This was speculated to be due to the suppression of genes encoding the production of the HDC enzyme, and therefore the precise mechanisms need to be further elucidated [58].

Modification of atmospheric conditions using gases of O_2_ and CO_2_ has also been studied to suppress the *M. psychrotolerans* and *P. phosphoreum* growth in fresh tuna. For safety reasons, this method can probably be used to replace the fresh tuna traditional vacuum packaging [103]. Some other alternative methods to control histamine accumulation in seafood products such as the application of potassium sorbate, sodium diacetate, high hydrostatic pressure treatment, and gamma irradiation have also been proposed and found to be effective to reduce the threat of histamine poisoning from seafoods [104].

## 8. Conclusions

Histamine is a colorless and odorless biogenic compound that is often unperceivable by consumers before poisoning occurs. Histamine-producing bacteria are responsible for producing this toxic biogenic amine and are commonly found in seafood products or their derivatives. This metabolite is produced by several bacterial species and is dominated by Gram-negative bacteria in seafood poisoning. This review highlighted the key role of these bacteria in histamine production in seafood and fish derivative products that cause human food poisoning following the consumption of these histamine-contaminated seafoods. Therefore, this review advances an integrated approach to our understanding of histamine poisoning as a health effect and elaborate the role of marine bacteria in the formation of histamine in seafood products. The article also highlights some methods to reduce or eliminate histamine production in seafood products, so that the associated poisoning following seafood product consumption can be avoided with a foundation for improved food safety.

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
