# Peer review of "Role of Marine Bacterial Contaminants in Histamine Formation in Seafood Products: A Review"

_microorganisms, 2022, doi:10.3390/microorganisms10061197_

Round 1
Reviewer 1 Report
the review is well written. the references are enough and in accordance with the manuscript. I suggest reviewing some words, which I highlighted in yellow in the PDF file. I believethey are errors of distraction also due to words repeated several times. In my opinion , the manuscript can be published after removing the errors.

Author Response
Point 1: The review is well written. The references are enough and in accordance with the manuscript. I suggest reviewing some words, which I highlighted in yellow in the PDF file. I believe
they are errors of distraction also due to words repeated several times. In my opinion, the manuscript can be published after removing the errors.
Response 1:
Thank you for your perception and correction on our manuscript. I have made some changes on the manuscript as per suggestion by reviewer 1. The revisions we made are highlighted with green.
Reviewer 2 Report
The authors aimed in this review paper to review the literature on Role of Marine Bacterial Contaminants in Histamine Formation in Seafood Products.
Unfortunately, the manuscript is of low quality in a scientific and technical way. The structure of the manuscript is quite confusing and some paragraphs and subsections are not related to the title and aim of the paper. Some basic rules, such as nomenclature, and way of writing of bacteria species were not followed, as well as reference styče required. Specific comments are given directly in the text.

Author Response
Point 1: Unfortunately, the manuscript is of low quality in a scientific and technical way. The structure of the manuscript is quite confusing and some paragraphs and subsections are not related to the title and aim of the paper. Some basic rules, such as nomenclature, and way of writing of bacteria species were not followed, as well as reference style required. Specific comments are given directly in the text.
Response 1: Thank you for your constructive criticisms on our manuscript. We made revisions to the manuscript based on Reviewer 2 suggestions so that our manuscript will meet the standard requirements of the journal. The revisions we make are highlighted with grey color.
Point 2: Line 69, Histamine responses in the human body. Is this sub section connected to the title?
Response 2: Thank you for your criticisms. Before we come to the actual role of marine bacteria in the histamine production, we wanted to inform the readers the health relevance and serious effects of histamine for human body. Later, when the readers get informed with this perspective, the details of how marine microorganisms play important role to produce such biogenic agent is key focus. In the last paragraph we inform the audience with methods to control histamine production in seafood products. Therefore from our point of view, this sub section is still related to the title of the manuscript. No changes have been made to this sub section.
Point 3: All genus and Species names have to be italicized
Response 3:
Thank you for your precise observation. We have made some revision in the manuscript and responding to reviewer 2 comments. The revisions are highlighted with grey.
In our manuscript (PDF version), all species names are italicized.
Point 4: Mahi-mahi fish (not everybody knows this fish)
Response 4: Thank you for your suggestion. We added the species name of mahi-mahi fish, so that all readers (scientists) are informed. This revision is highlighted with grey color.
Point 5: The reviewer concern about the use of copper in water (line 251-253)
Response 5: Thank you for your observation. In these sentences, we want to inform that copper has recently been found to contaminate water body (sea water). This contaminant has been a great concern worldwide. In other words, pollution in aquaculture areas may negatively impact edible species and threaten seafood quality and safety. Copper exposure increased its concentration in crayfish meat by 17.4%, but the copper concentration remained within acceptable food safety limits (Soedarini et al., 2014). Soedarini et al. (2014) further reported that elevated copper levels affected the bacterial community both in the water used to cultivate crayfish and in the marbled crayfish themselves.
They came to the conclusion that copper from the habitat appears to reduce histamine accumulation in crayfish meat during storage by affecting the bacterial community structure of the cultivation water and most likely also in the intestine of the crayfish. From a food safety point of view, copper treatment during the aqua culturing of crustaceans has a positive impact on the postharvest stage.
In our manuscript, we do not propose to use copper as preservative agent to reduce histamine level in the seafood products. We only inform that copper pollution significantly affect the community of histamine producing bacteria in the seafood product following copper exposure that may be found in the sea water. We do not make any changes in this part of the manuscript.
Point 6: Line 259 –270: The reviewer ask about the relationship between this paragraph and the title, whether they are related each other.
Response 6:
Thank you for your observation. We think that this paragraph is closely related to the title of our manuscript. Histamine producing bacteria can easily contaminate seafood products and their density will increase when in contact with free histidine of the seafood product. In this occasion, they will activate their gene encoding histamine decarboxylase enzyme, so they can utilize free histidine as a source of energy.
In this paragraph, we try to elaborate how the histamine is produced by Gram positive and Gram negative bacteria. The mechanism of transport of histamine from bacterial cytoplasm is also briefly described. Histamine exported from bacterial cytoplasm will flood the fish muscle and accumulate in the seafood product. We did not make any changes on this part.
Point 7: Lines 332 – 345: the reviewer could not see the relationship between this paragraph with the title of section 4.
Response 7: Thank you for your criticism. The title of section 4 is detection of histidine decarboxylase cluster (HDC) gene.
In the previous paragraph in this section 4, we mention several genes of HPB (histamine producing bacteria) involved in the histamine formation. When these genes become active, histamine will be detected in the seafood products. In the paragraph written on lines 332 – 345, we elaborate some methods that have been used to measure histamine level as an indication of expression of genes involved in the histamine formation by HPB. From this point of view, we think this paragraph is still related to the title of section 4.
Point 8: Line 417: The reviewer found that Anti-Histamine compounds and solutions is not correlated with the aim of the review.
Response 8: Thank you for your observation and criticism. This section tries to explain some solutions when food poisoning happens in the community following consumption of histamine-contaminated seafood products.
The aim of this review is to provide people with information on the role of seafood products as a vehicle for histamine produced by HPB to reach human or animals and cause poisoning. When this happens, people have alternatives (solutions) to avoid the fatality of histamine poisoning.
Therefore, (from our point of view) we think that this section is still relevant to be included as part of our review paper.
Point 9: Reference style must be revised
Response 9: Thank you for your observation and suggestion. We have made revision on the references accordingly. They are grey highlighted.
Reviewer 3 Report
The topic of the article: „Role of Marine Bacterial Contaminants in Histamine Formation in Seafood Products: A Review” falls within the thematic scope of the journal MICROORGANISMS.
I have no objection to the purposefulness of the selected literature or to the general organization of the manuscript. The division into individual chapters is correct and covers the whole of the discussed problem.
My first remark concerns the lack of any purpose of the article at the end of the "Introduction" chapter - 1-2 sentences.
The second remark concerns the last chapter "Summary". In my opinion, there are no 1-2 sentences discussing the gaps in the current state of knowledge. There is no presentation of what research directions according to the authors are poorly known. Typically, summaries of this type are found in review articles.
The following is also repeated throughout the manuscript:
- no use of abbreviations previously introduced by the authors, - no italics are used for the names of genera, species of microorganisms and names of genes,
- large gaps in bibliographic data in the References chapter.
All comments and suggestions for corrections were introduced in the review mode to the attached pdf file.

Author Response
Point 1: The topic of the article: “Role of Marine Bacterial Contaminants in Histamine Formation in Seafood Products: A Review” falls within the thematic scope of the journal MICROORGANISMS.
Response 1: Thank you for your positive comments on our manuscript.
Point 2: I have no objection to the purposefulness of the selected literature or to the general organization of the manuscript. The division into individual chapters is correct and covers the whole of the discussed problem.
Response 2: Thank you for your interest and positive comments on the overall manuscript.
Point 3: My first remark concerns the lack of any purpose of the article at the end of the "Introduction" chapter - 1-2 sentences.
Response 3: Thank you for your suggestions. We added two sentences containing the purpose of the article. The sentences are highlighted in yellow.
Point 4: Line 79 DCs (The abbreviation is no explained)
Response 4: Thank you for your preciseness. We add what DCs stand for in the manuscript. It is highlighted in yellow.
Point 5: Line 116 (remove histidine decarboxylase)
Response 5: We remove the words of histidine decarboxylase from the manuscript.
Point 6: Genus and species names (156 – 168) need to be italicized)
Response 6: Thank you for your suggestion. We have made revisions per suggested by reviewer and highlighted with yellow (grey in some cases as also suggested by other reviewer) color in the manuscript.
Point 7: The second remark concerns the last chapter "Summary". In my opinion, there are no 1-2 sentences discussing the gaps in the current state of knowledge. There is no presentation of what research directions according to the authors are poorly known. Typically, summaries of this type are found in review articles.
Response 7: Thank you for your criticisms. The key focus of this review is how marine microorganisms play important role to produce histamine and the mechanisms from a health and food safety point of view. This has been integrated with understanding of actual role of marine bacteria in the histamine production. The readers are further informed with an integrated understanding of the health relevance and serious effects of histamine for human body. Subsequently, when the readers get informed with this perspective the audiences are introduced to the methods to control histamine production in seafood products. Therefore, this integrative approach to understanding the food safety challenge and associated health effects of histamine is the strength of this review. We added two sentences containing the purpose of the article. The sentences are highlighted in yellow.
Point 8: No use of abbreviations previously introduced by the authors, - no italics are used for the names of genera, species of microorganisms and names of genes,
Response 8: Thank you for your observation. We have made all corrections per suggested by the reviewer 3 and they are highlighted in yellow (or grey in some cases as other reviewer also has the same comments).
Point 9: Large gaps in bibliographic data in the References chapter.
Response 9: Thank you for your observation. We have revised our manuscript accordingly and highlighted with grey color in this section of the manuscript).